# Executive Function Response to Moderate-to-High-Intensity Rope Skipping in Overweight Adolescents Aged 12–14: A Preliminary Study

**DOI:** 10.3390/jfmk10020152

**Published:** 2025-04-29

**Authors:** Qian Yu, Xiaodong Wang, Lin Zhang

**Affiliations:** Physical Education School, Shenzhen University, Shenzhen 518060, China; 18892691501@163.com (Q.Y.); wang_xd@szu.edu.cn (X.W.)

**Keywords:** rope skipping, overweight adolescents, obese adolescents, executive function

## Abstract

**Background**: Previous studies have shown that a high body mass index (BMI) is detrimental to executive function (EF) in children and elderly populations but may be improved by physical activity (PA). However, research on adolescents aged 12–14 is lacking. This study explores two parts: Part 1: cross-sectional correlation between BMI and EF; Part 2: the effect of an 8-week rope skipping intervention. **Methods**: Recruit 166 adolescents to participate in EF tasks. Screen and select 96 adolescents to be assigned to the normal weight control group (N-C, *n* = 23) and the normal weight exercise group (N-E, *n* = 23), the overweight control group (OV-C, *n* = 13), and the overweight exercise group (OV-E, *n* = 14), the obese control group (OB-C, *n* = 12), and the obese exercise group (OB-E, *n* = 11). Exercise program: moderate-to-high-intensity rope skipping training. Control program: Self-directed reading. Data were recorded for BMI and EF tasks. **Results**: Part 1, EF and BMI exhibit a negative linear correlation. Part 2, the reaction time of EF tasks in N-E, OV-E and OB-E decreased about 9, 14, 13% (*p* < 0.05), respectively, while the average BMI in OV-E and OB-E decreased about 10% and 11% (*p* < 0.05), suggesting a potential correlation between the reduction in BMI and the improvement in EF, which means that the exercise intervention significantly improved EF performance. **Conclusions**: Among adolescents aged 12–14, overweight and obese individuals exhibit weaker EF compared with normal weight individuals. An 8-week moderate-to-high-intensity rope skipping exercise program can improve EF in adolescents aged 12–14 with different BMIs, and the improvement is greater in overweight and obese individuals.

## 1. Introduction

Based on recent trends, it is projected that by 2035, over 750 million children aged between 5 and 19 years will be afflicted with overweight/obesity, as gauged by body mass index (BMI, kg/m^2^) [1]. In 2019, the overall prevalence of overweight/obesity among children and adolescents in China was 23.4%, with obesity at 9.6% [2]. Obesity is a crucial pathogenic factor for a multitude of chronic diseases, including cardiovascular disease, type II diabetes, coronary heart disease, chronic kidney disease, and site-specific cancers [3]. Obesity during childhood can endure into adulthood, predisposing individuals to increased risk of non-communicable diseases in adulthood [4]. Executive function (EF) consists of higher-order cognitive abilities that enable conscious control of thoughts, actions, and emotions for goal-directed behavior and adaptive functioning [5]. Increasing evidence suggests that obesity in adults or children may negatively affect brain health, specifically brain development [6] and EF [7], mainly reflected in the reduction in inhibitory control, working memory, and cognitive flexibility [8]. It is hypothesized that such a reduction could be attributed to alterations in the brain’s microstructure, such as white matter hyperintensity (WMH), brain atrophy, and cerebral microbleed [9].

Regular physical activity (PA) has received significant attention in recent years due to its accessible, flexible, cost-effective, scalable, and well-documented health benefits [10], which encompass enhancements in brain tissue volume, preservation of white matter microstructural integrity, mitigation of WMH, prevention of cerebral small vessel disease, and improvement in cardiometabolic health [11,12,13]. Functionally, the prefrontal cortex (PFC) plays a part in distractor-resistant maintenance, while updating is thought to be supported by the striatum, and the neurotransmitter dopamine plays a crucial role in balancing these complementary processes [14]. Notably, exercise increases the release of dopamine [15], which, in turn, is intricately linked to the enhancement of the EF [16], as the previously discussed physiological factors are positively correlated with PA. This suggests that exercise may enhance and preserve the functioning of these mechanisms, thereby improving EF. Compared to the overweight/obese population, the relative weakness of these mechanisms provides a critical target for achieving more pronounced intervention effects.

Aerobic exercise targets cardiorespiratory fitness and mitochondrial biogenesis [17]. Interval training is characterized by brief bursts of intense effort interspersed by short periods (intervals) of recovery [18]. Both high-intensity cardiovascular exercise and chronic exercise interventions might be a promising way to promote multiple aspects of EF in children and adolescents [19,20]. Frequent participation in tennis play is related to better EF in children and adolescents [21]. One study found that coordination training significantly improved children’s attention than training focused on endurance, strength, and cardiovascular health [22]. Rope skipping is a cost-effective strategy that increases cardiovascular health and reduces cardiometabolic risk factors and BMI in young populations [23]. Liu et al. (2024) found that acute rope skipping can improve cognitive function in children [24]. Additionally, a comprehensive meta-analysis concluded that both high- and moderate-intensity exercise regimens effectively improve EF [20]. However, there is a lack of research on the use of moderate-to-high-intensity exercise for interval training with overweight/obese adolescents aged between 12 and 14 as participants to assess the effect on EF. The tripartite model of EF, which includes distinct components such as inhibitory, working memory, and cognitive flexibility, typically emerges during early adolescence, around the ages of 13 to 14 years [25].

This study conducted two evaluations. Part one: We aimed to identify the association between BMI and EF in adolescents. We hypothesized that there would be a negative correlation between higher BMI and poorer cognitive performance in adolescents. Part two: We aimed to identify and investigate the effect of an 8-week moderate-to-high-intensity rope skipping training program and whether it improved EF in the overweight/obese adolescents’ group. It was hypothesized that EF would be improved by rope skipping exercise intervention in overweight/obese adolescents.

## 2. Materials and Methods

### 2.1. Study Design and Participants

All participants/parents provided informed consent prior to enrolment. The study was conducted at a single secondary school in Shenzhen, China.

The sample size was determined by power analysis using G-Power software (version 3.1) [26]. The first study utilized values of α = 0.05, power = 0.80, effect size = 0.25, with three groups (normal weight, overweight, and obese) and two measurements (BMI and EF) in a priori type of power analysis [27]. The minimum sample size required was *n* = 82. The second study utilized values of α = 0.05, power = 0.80, effect size = 0.25, with six groups (normal weight control and exercise groups, overweight control and exercise groups, obese control and exercise groups) and two measurements (BMI and EF) in a priori type of power analysis [28]. The minimum sample size required was *n* = 66. Each group requires at least 22 participants.

A total of 166 subjects aged between 12 and 14 years participated in the correlational study, out of which 96 agreed to continue their involvement in the exercise intervention program. Inclusion criteria were as follows: (a) Adolescents 12–14 years old whose BMI was within normal weight guidelines, or those who were overweight or obese as defined by age and gender [29]; (b) without cardiovascular, neurological, psychiatric disease, or serious injuries (such as external/internal traumatic injuries) in the last six months; (c) no hormones, weight loss, or psychotropic drugs had been consumed in the last six months. All participants signed informed consent forms. The baseline characteristics of the participants are summarized in Table 1.

### 2.2. Study Intervention

Exercise intervention: an 8-week moderate-to-high-intensity rope skipping training (3 times per week). Each session included a general warm-up (dynamic exercises), a moderate-to-high-intensity rope skipping training intervention, and a cool-down period (static stretching exercises) for a total of 50 min (Table 2). The rope skipping training consisted of fast jumping with feet together. The sedentary control groups maintained their resting heart rate during the training time without any prescribed PA. Control intervention: Sit quietly and read, ensuring that your heart rate does not exceed 100 beats per minute. Specific movement demonstrations are provided in Appendix A.

During the intervention, telemetric heart rate measurement was recorded using a fitness smart band (SHB-02, Shape, Shenzhen, China).

### 2.3. Study Outcomes

EF encompasses three domains: inhibitory control (cognitive inhibitory and behavioral inhibitory), working memory, and cognitive flexibility, which were assessed by the Flanker task, Stop-Signal Task (SST) [30], 2-Back task [31] and More-odd Shifting (MOS) task [32], respectively. Participants completed the tasks on E-prime 3.0 software. Body weight (kg) was measured with a smart body fat scale (HUAWEI, model: HAG-B19, Shenzhen, China). Height (m) was measured with a stadiometer (Shuang sheng, Danyang, China). BMI was calculated as weight in kilograms divided by the square of height in meters.

#### 2.3.1. Flanker Task

A modified version of the Eriksen flanker test was used to assess cognitive inhibitory [33]. The task consists of congruent and incongruent parts. The incongruent situation (for example: →→←→→/←←→←←); the congruent situation (for example: ←←←←←/→→→→→). For the congruent situation, the participant must press “F” on the keyboard; otherwise, they must press “J.” Participants were required to respond within 1500 milliseconds (ms); if not, it was recorded as no response. There were 96 tests in the formal experiment, and the congruent and incongruent stimuli were presented 48 times each (Figure 1A). The difference in times to complete the congruent and incongruent tasks and the average of both accuracies were used to reflect the cognitive inhibitory ability.

#### 2.3.2. SST

A modified version of the SST was used to assess behavioral inhibitory [30]. Participants were required to press the corresponding “Q” key on the keyboard when a black circle arrow pointing to the left appeared. If a right-pointing arrow appeared, participants were required to press the “P” key. When a red circle arrow pointing to the left or right appeared on the screen, participants were not required to respond. The stimulus was presented for 500 ms—exceeding this range was recorded as no response. In total, there were 150 tests in the formal experiment. The reaction time and accuracy of the SST were used to reflect the behavioral inhibitory ability (Figure 1B).

#### 2.3.3. 2-Back Task

A modified version of the 2-Back task was used to assess working memory [31]. In this task, a number appeared in the center of the screen. Participants were required to start from the third number and assess whether it was the same as the first, and then they were to move to the fourth number and assess whether it was the same as the second, and so on. Participants pressed the “F” key for consistency and the “J” key for inconsistency. The stimulus was presented for 3000 ms, and beyond this range was recorded as no response. There was a total of 48 tests in the formal experiment (Figure 1C). The reaction time and accuracy of the 2-Back task were used to assess working memory.

#### 2.3.4. MOS Task

A modified version of the MOS task was used to assess cognitive flexibility [34]. The task consisted of unshifting (judging large/small or odd/even separately) and shifting parts (judging conversion of large/small and odd/even). Some numbers (1–9) appeared in the MOS task screen, but there was no 5. Large/small judgments (black numbers): for numbers ˂5, participants had to press “F”; for numbers ˃5, they were required to press “J.” Odd/even judgments (green numbers): for odd numbers, participants had to press “F”; for even numbers, participants had to press “J.” Conversion of large/small and odd/even judgments: if the presented number was black, participants had to make a large/small judgment; if the presented number was green, they had to make an odd/even judgment. The stimulus presentation had no time limit, and the subsequent test was performed after the participant answered. The formal experiments of the three parts consisted of 48 tests (Figure 1D). The accuracy and the differences in times to complete unshifting and shifting tasks were used to reflect cognitive flexibility.

### 2.4. Quality Monitoring

We utilized the intelligent sports classroom exercise load guidance system fitness trackers (Appendix A) to monitor the average heart rate during each session, ensuring the effective operation of the experimental data. Adherence to the exercise intervention was measured by the average heart rate during the moderate-to-high jump rope training sessions out of the 24 scheduled sessions [35]. Compliance with the intervention was assessed by calculating each participant’s average heart rate during the moderate-to-high jump rope training sessions and comparing it to the target heart rate. Data were deemed invalid if the participant’s average heart rate fell below the target heart rate. In the context of dietary intervention, health and nutrition intake workshops were conducted for the participants, and corresponding healthy recipes were provided to their parents.

### 2.5. Statistical Analysis

Data are shown as mean ± SD. SPSS (version 27.3) and GraphPad Prism software (version 10.3) were used for data analysis. Part 1: The Shapiro–Wilk test was used to verify normal data distribution. Spearman’s correlation was used to determine correlations between BMI and EF. Correlations where *p* < 0.05 were selected to construct the linear graphs. One-way ANOVA (Analysis of Variance) was used to examine the differences in EF tasks among the normal weight group, the overweight group, and the obese group. Part 2: Two-way ANOVA with post hoc analysis. For BMI x Intervention, analyze the differences between groups after the intervention. The Bonferroni correction was used for pairwise comparisons. Differences were considered significant when *p* <0.05.

## 3. Results

Part one: Two hundred students are within the scope of recruitment. Twenty-three students with a low BMI and a thinner and leaner stature did not meet the criteria and were excluded. Eleven students refused to participate in the experiment due to personal time constraints. This resulted in one hundred and sixty-six participants being retained. Part two: Ninety-six participants were involved in the experiment. Seven students lost contact during the intervention; six students discontinued the intervention due to being busy. This resulted in eighty-three participants being retained (Figure 2).

### 3.1. Correlation Between BMI and EF in Adolescents

There was a significant linear positive correlation between BMI and the RT of the Flanker task (*r* = 0.6614, *p* < 0.0001, Figure 3A), SST (*r* = 0.5711, *p* < 0.0001, Figure 3C), 2-Back task (*r* = 0.5038, *p* < 0.0001, Figure 3E), MOS task (*r* = 0.5283, *p* < 0.0001, Figure 3G), the accuracy of the Flanker task (*r* = 0.1714, *p* = 0.0272, Figure 3B), 2-Back task (*r* = −0.3347, *p* < 0.0001, Figure 3F), and MOS task (*r* = −0.2680, *p* = 0.0005, Figure 3H). This result indicates that a higher BMI leads to a longer response time in EF tasks; therefore, participants with a higher BMI exhibited lower accuracy in the 2-Back and MOS tasks. Higher BMI was significantly associated with longer reaction times across all EF tasks (Flanker, SST, 2-Back, MOS; all *r* > 0.50, *p* < 0.0001) and reduced accuracy in working memory and cognitive flexibility tasks (2-Back/MOS; *r* = −0.27 to −0.33, *p* ≤ 0.0005), indicating higher BMI is linked to slower thinking speed and lower accuracy.

### 3.2. Descriptive Statistics and Comparative Analysis of Pre-Exercise Intervention

Since no significant within-group gender differences were observed (all *p* > 0.05), data were aggregated across genders within each experimental group for subsequent analyses. As shown in Table 3, there were significant differences in EF among different BMI groups (*p* < 0.0001). According to the reaction time data from the EF tasks, higher BMI levels were associated with relatively longer reaction times. Additionally, some tasks exhibit a relatively low accuracy rate.

### 3.3. Effects of Moderate-to-High-Intensity Rope Skipping on BMI in Normal Weight and Overweight/Obese Adolescents

As shown in Figure 4A, based on BMI groupings, we categorized the intervention methods into N-C, N-E, OV-C, OV-E, OB-C, and OB-E. Significant differences were observed between OV-C and OV-E (*p* < 0.0001), OB-C and OB-E (*p* < 0.0001). The BMIs of the OV-E and the OB-E groups were nearly 9% and 11%, respectively, a reduction from pre-exercise intervention.

### 3.4. Effects of Moderate-to-High-Intensity Rope Skipping on EF in Normal Weight and Overweight/Obese Adolescents

Before exercise intervention, there were no significant differences between N-C and N-E, OV-C and OV-E, OB-C, and OB-E in EF tasks. After 8 weeks of exercise, we will compare the performance of EF tasks following the post-exercise intervention.

#### 3.4.1. Effects of Moderate-to-High-Intensity Rope Skipping on Cognitive and Behavioral Inhibitory in Normal Weight and Overweight/Obese Adolescents

As shown in Table 4, ANOVA revealed significant main effects for BMI groups and intervention methods (exercise vs. control) on reaction times in the Flanker task (*p* < 0.0001, *p* < 0.0001) as shown in Figure 4B,C. A significant interaction was also observed between BMI groups and intervention methods (*p* = 0.019). Post hoc analyses with Bonferroni corrections demonstrated that the exercise group showed significantly greater improvement in cognitive inhibitory compared to the control group across normal weight, overweight, and obese groups (*p* = 0.004, *p* < 0.0001, *p* < 0.0001), was sequentially shortened close to 25, 35, 36% after the intervention. However, no significant main effects were observed in accuracy rates, indicating that, in the absence of significant differences in accuracy, the exercise group exhibited shorter reaction times compared to the control group. This suggests that exercise intervention has a beneficial impact on cognitive inhibitory control among adolescents of normal weight, overweight, and obese individuals.

As is shown in Table 4, Figure 4D,G, the ANOVA revealed significant main effects for BMI groups and intervention methods on reaction times in the SST (*p* < 0.0001, *p* < 0.0001), and a significant interaction between BMI groups and intervention methods (*p* = 0.049). Post hoc analyses with Bonferroni corrections demonstrated that the exercise group showed significantly greater improvement compared to the control group across the overweight and obese groups (*p* = 0.002, *p* = 0.001), and was sequentially shortened by about 7% and 8% after the intervention. Although no significance was observed in N-E, it still showed a 5% reduction compared to pre-intervention levels. Regarding accuracy, a significant main effect was observed for the intervention, but no significant interaction effects were found. This result shows that exercise has a more significant effect on improving behavioral inhibitory control in overweight and obese children compared to the normal-weight adolescents.

#### 3.4.2. Effects of Moderate-to-High-Intensity Rope Skipping on Working Memory in Normal Weight and Overweight/Obese Adolescents

As presented in Table 4, Figure 4E,H, the ANOVA results indicated significant effects on both BMI groups and intervention methods regarding reaction times in the 2-Back task (*p* < 0.0001, *p* = 0.003). However, no significant interaction was observed between BMI groups and intervention methods. When we conducted post hoc analyses with intervention methods as the main effect, significant differences were found only in the OV-E (*p* = 0.027), which was shortened by close to 9%, while the OB-E approached a significance of (*p* = 0.075) and was shortened by close to 10%. In terms of accuracy, differences were observed in the analysis with BMI groups as the main effect, but no interaction effects were present. Post hoc analyses also revealed no significant differences within each group. This finding indicates that exercise exerts a more pronounced impact on enhancing working memory abilities in overweight and obese children than in the normal-weight adolescents.

#### 3.4.3. Effects of Moderate-to-High-Intensity Rope Skipping on Cognitive Flexibility in Normal Weight and Overweight/Obese Adolescents

The results of the ANOVA show that the RT of the MOS has a significant main effect for both BMI groups and intervention methods (*p* < 0.0001, *p* < 0.0001), but no significant interaction effects between BMI groups and intervention methods (Table 4, Figure 4F,I). Post hoc analyses conducted separately for each main effect revealed that the exercise group showed significantly greater improvement in EF compared to the control group within the normal weight, overweight, and obese groups (*p* = 0.016, *p* < 0.0001, *p* = 0.012), was shortened nearly 19, 24, 21% after the intervention, respectively. In terms of accuracy, significance was only observed in the ANOVA with exercise as the main effect. Post hoc analyses revealed that the exercise group showed significantly greater improvement in EF compared to the control group within the normal weight, overweight, and obese groups (*p* = 0.003, *p* = 0.007, *p* = 0.011). Based on the results, we can conclude that exercise can significantly improve cognitive flexibility in adolescents of normal weight, overweight, and obese individuals. However, the extent of improvement is not significantly correlated with BMI levels.

## 4. Discussion

As it was hypothesized in this study, the first evaluation (Part 1) explored the association between BMI and EF, revealing that higher BMI is linked to lower EF. Additionally, the second evaluation (Part 2) investigated the effects of an 8-week moderate-to-high-intensity rope skipping program on EF. The results showed that rope skipping effectively improved inhibitory control and cognitive flexibility in overweight and obese adolescents, with similar improvements observed in normal-weight children. Notably, the 8-week program also significantly enhanced working memory in adolescents aged 12–14, particularly in the overweight group.

Inhibitory control includes cognitive and behavioral aspects: cognitive inhibitory suppresses distractions to focus on tasks, while behavioral inhibitory restrains impulsive actions [5,36]. Working memory has limited capacity and involves interactions between cognitive processing, sensorimotor mechanisms, and long-term memory activation [37,38]. Cognitive flexibility relates to the ability to shift between tasks or mental sets [39].

In part one, we analyzed the correlation between EF performance—measured by reaction time and accuracy—and BMI in a sample of 166 participants. The results revealed no significant correlation between BMI and accuracy in the Flanker task (cognitive inhibitory control) or the SST (behavioral inhibitory control). However, reaction times were positively correlated with BMI, indicating that individuals with higher BMI exhibited slower response times. The lack of association between BMI and accuracy suggests that task performance, in terms of correctness, remained unaffected, while variations in reaction time directly reflected differences in inhibitory control. For the 2-Back (working memory) and MOS (cognitive flexibility) tasks, significant BMI-related correlations were observed. Specifically, higher BMI was associated with lower accuracy and longer reaction times, suggesting that individuals with higher BMI demonstrated weaker working memory and reduced cognitive flexibility. These findings are consistent with prior systematic reviews, wherein eight of nine cross-sectional studies reported significantly impaired cognitive performance in obese children/adolescents relative to normal-weight controls [40].It was also observed in older populations, where obesity has been linked to memory impairments in elderly individuals [41]. Mechanistic studies suggest that obesity is associated with structural and functional brain alterations, including cortical thinning, reduced brain volume, and abnormalities in key regions such as WMH, gray matter, the hippocampus, the hypothalamus, and the left dorsolateral prefrontal cortex [42,43,44,45,46]—all of which play critical roles in EF development and regulation. Furthermore, obesity-induced systemic inflammation may contribute to neuroinflammation [47], potentially accelerating cognitive decline.

An eight-week high-intensity rope skipping study examined its effects on the Flanker task (cognitive inhibitory control) and SST (behavioral inhibitory control) in exercise and control groups. While no significant differences in accuracy were observed among overweight, obese, and normal-weight groups (*p* > 0.05), reaction times improved significantly following the intervention. In the Flanker task, the exercise group exhibited faster reaction times across all weight categories: overweight (*p* < 0.0001), obese (*p* < 0.0001), and normal-weight groups (*p* = 0.004). For the SST, significant improvements were observed exclusively in the overweight (*p* = 0.002) and obese groups (*p* = 0.001), with reduced task completion times. These findings suggest that high-intensity exercise enhances both cognitive and behavioral inhibitory control efficiency without affecting accuracy, particularly benefiting overweight and obese adolescents. Improvements in cognitive inhibitory control, as measured by the Flanker task, were consistent across all weight groups, underscoring the universal cognitive benefits of exercise. However, the overweight and obese groups, starting from lower baseline performance levels, demonstrated greater relative improvements (35% in overweight, 36% in obese, and 25% in normal-weight participants), highlighting the amplified benefits of exercise for these populations. In contrast, enhancements in behavioral inhibitory control were weight-specific, with significant improvements observed only in overweight and obese participants. Previous electroencephalography (EEG) studies have reported that exercise reduced current source density in the mid-frontal gyrus, indicating enhanced neural efficiency and supporting improved inhibitory control performance [48]. This neural mechanism provides a physiological basis for the observed cognitive benefits. Additionally, the significant BMI reductions in overweight and obese participants post-intervention may further contribute to the observed improvements in inhibitory control performance.

An eight-week high-intensity rope-skipping intervention assessed the effects of exercise on working memory using the 2-Back task in exercise and control groups. While no significant differences in task accuracy were observed among overweight, obese, and normal-weight participants (*p* > 0.05), reaction times improved significantly in the OV-E (*p* < 0.05) and showed a trend toward improvement in the OB-E (*p* = 0.075), indicating enhanced task efficiency. These findings suggest that exercise improves working memory performance by increasing processing speed without compromising accuracy. No significant improvement was observed in the N-E (*p* = 0.206), indicating that the cognitive benefits of exercise may be group-specific, primarily favoring overweight and obese individuals who started with lower baseline performance. This aligns with previous studies by de Bruijn et al., Meijer et al., and Wassenaar et al., which reported positive but non-significant effects of exercise on working memory in normal-weight adolescents [49,50,51]. The limited improvement in this group may be attributed to their higher baseline working memory capacity, resulting in smaller marginal gains, or to the possibility that working memory is less sensitive to short-term exercise interventions and may require longer training durations to elicit measurable effects.

An eight-week high-intensity rope-skipping intervention was conducted to evaluate its effects on cognitive flexibility using the MOS task. Compared to the control group, the exercise group demonstrated significant improvements in accuracy across all weight categories: overweight (*p* = 0.007), obese (*p* = 0.025), and normal weight (*p* = 0.003), indicating that exercise effectively enhances cognitive flexibility accuracy. Additionally, reaction times improved significantly in all groups (*p* = 0.001, *p* = 0.012, *p* = 0.016), reflecting faster task completion. The consistent improvements observed across all weight groups highlight the universal cognitive benefits of exercise. These findings align with previous studies, such as those by Ortega et al., which demonstrated that exercise enhances cognitive function and brain health in overweight and obese children by improving brain structure (e.g., increased gray matter volume) and neural connectivity [52]. Similarly, Costigan SA et al. found that HIIT positively impacted cognitive function and mental health in adolescents [53]. However, Wassenaar TM et al. reported that while a one-year high-intensity exercise intervention improved fitness and some cognitive functions in adolescents, cognitive flexibility did not significantly improve [51]. These differences may arise from varied cognitive tasks and exercise intervention features, such as mode, frequency, duration, intensity, and length.

This study provides compelling evidence supporting the cognitive benefits of moderate-to-high-intensity rope skipping, with particularly pronounced effects observed in overweight and obese adolescents. The research makes significant contributions to exercise science, cognitive psychology, and pediatric health through its rigorous methodology, comprehensive assessment of EF, and practical intervention design.

However, several limitations should be noted. The relatively small sample size in the exercise intervention may affect the generalizability of the findings. Furthermore, the study did not account for potentially influential covariates, including dietary patterns, habits, sleep patterns, pubertal status, and baseline physical activity levels—may have influenced the observed results. For instance, variations in macronutrient intake or sleep duration could independently modulate cognitive performance [54,55]. Although participants were instructed to maintain their usual lifestyles during the trial, these factors were not systematically measured. The absence of long-term follow-up assessments represents another limitation, as it precludes evaluation of the intervention’s sustained effects. Additionally, the reliance on BMI rather than more precise measures like body fat percentage for classifying overweight/obesity status may impact the validity of the findings. Most importantly, the study did not investigate the underlying physiological and psychological mechanisms responsible for the observed cognitive improvements, which limits our understanding of how exercise influences executive function at a mechanistic level.

These limitations point to critical future research needs: larger samples, rigorous confounder control (via 24-h dietary recalls, actigraphy, and accelerometry), longer follow-ups, precise body composition analysis (e.g., DEXA), and mechanistic studies to unravel how rope skipping augments executive function in this population.

## 5. Conclusions

This study highlights that EF is weaker in overweight and obese adolescents compared to their normal-weight peers. However, an 8-week moderate-to-high-intensity rope skipping intervention effectively improved EF in 12–14-year-old adolescents with overweight/obesity, with concurrent BMI reductions observed in the overweight/obese groups. Future research should focus on designing personalized exercise programs tailored to age and health status to optimize cognitive benefits. Furthermore, investigations should elucidate BMI’s potential mediating role in exercise-induced EF improvements. Beyond phenomenological studies, mechanistic research should explore underlying pathways, including neuroplasticity, gut microbiota, and inflammatory responses. Longitudinal follow-up studies are also imperative to evaluate the sustained cognitive effects of exercise interventions and identify determinants of long-term efficacy.

## Figures and Tables

**Figure 1 jfmk-10-00152-f001:**
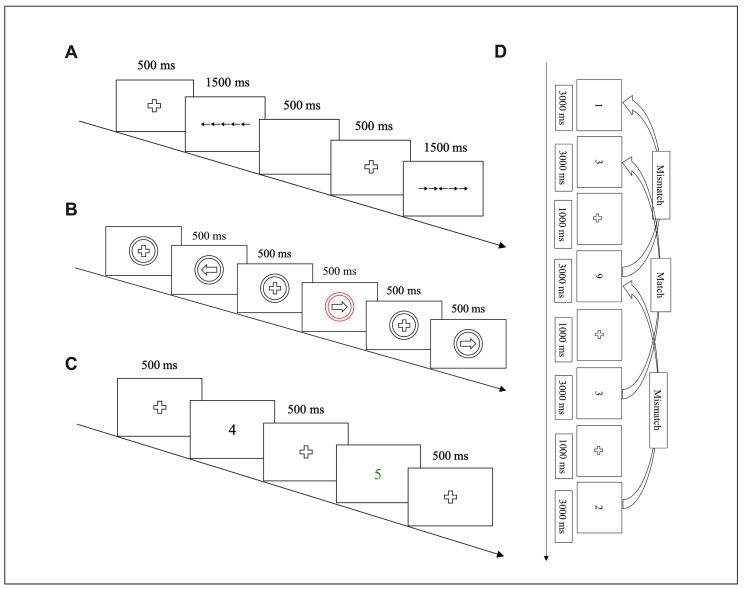
Flow diagram of the executive functional tasks. (**A**) Flow diagram of the Flanker task. (**B**) Flow diagram of the SST. (**C**) Flow diagram of the 2-Back task. (**D**) Flow diagram of the MOS task. 
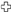
/
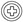
, fixation point; →→←→→/←←→←←, incongruent situation; ←←←←←/→→→→→, congruent situation; 
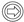
, right-pointing arrow; 
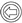
, left-pointing arrow; 
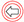
, red circle arrow pointing to the left; 
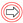
, red circle arrow pointing to the right; 4/5/1/3/9/3/2, numbers.

**Figure 2 jfmk-10-00152-f002:**
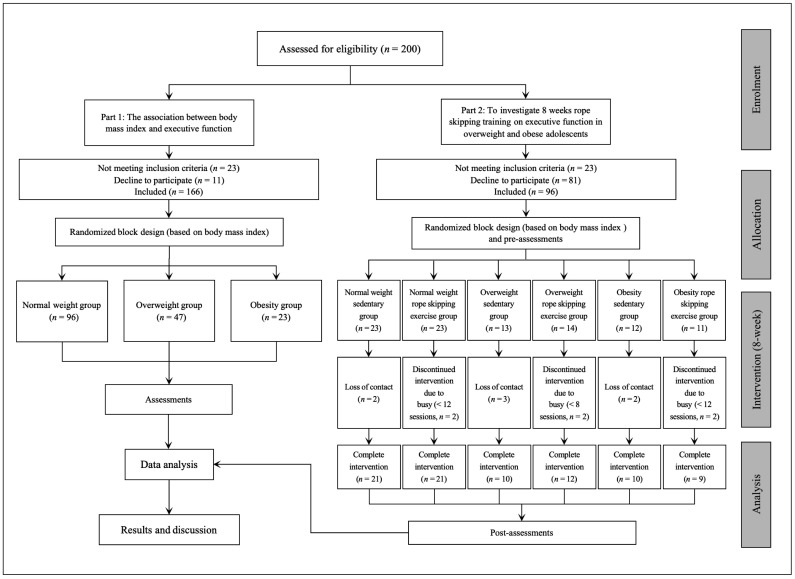
Participant selection flowchart of the study.

**Figure 3 jfmk-10-00152-f003:**
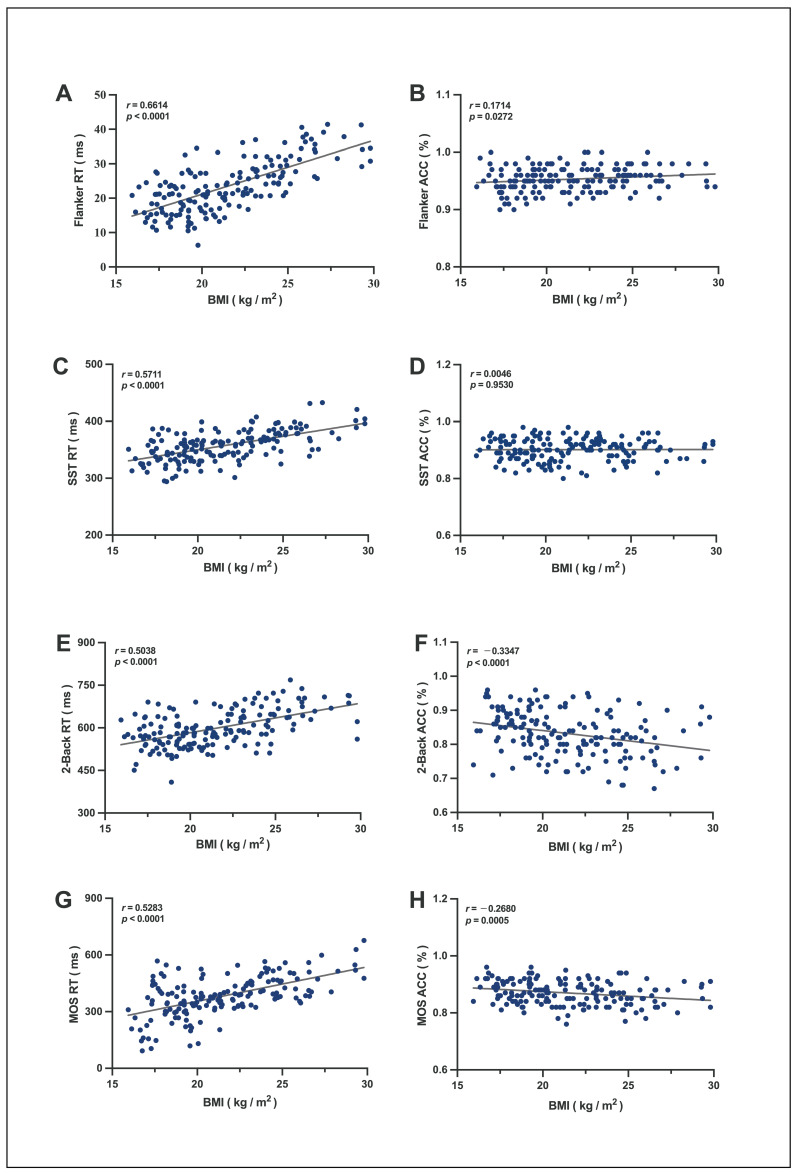
Linear plot of the correlation between BMI and EF. Correlation findings are shown as Spearman’s correlation coefficient (*p*-value); strong correlations with |*r*| ≥ 0.8, moderate correlations with 0.5 ≤ |*r*| < 0.8, weak correlations with 0.3 ≤ |*r*| < 0.5, and no correlations with |*r*| < 0.3. *p* < 0.05 are shown as a significant correlation. (**A**) Linear plot of the correlation between BMI and Flanker task reaction (ms). (**B**) Linear plot of the correlation between BMI and Flanker task accuracy (%). (**C**) Linear plot of the correlation between BMI and SST reaction (ms). (**D**) Linear plot of the correlation between BMI and SST accuracy (%). (**E**) Linear plot of the correlation between BMI and 2-Back reaction (ms). (**F**) Linear plot of the correlation between BMI and 2-Back accuracy (%). (**G**) Linear plot of the correlation between BMI and MOS reaction (ms). (**H**) Linear plot of the correlation between BMI and MOS accuracy (%). RT, reaction time; ACC, accuracy; and BMI, body mass index.

**Figure 4 jfmk-10-00152-f004:**
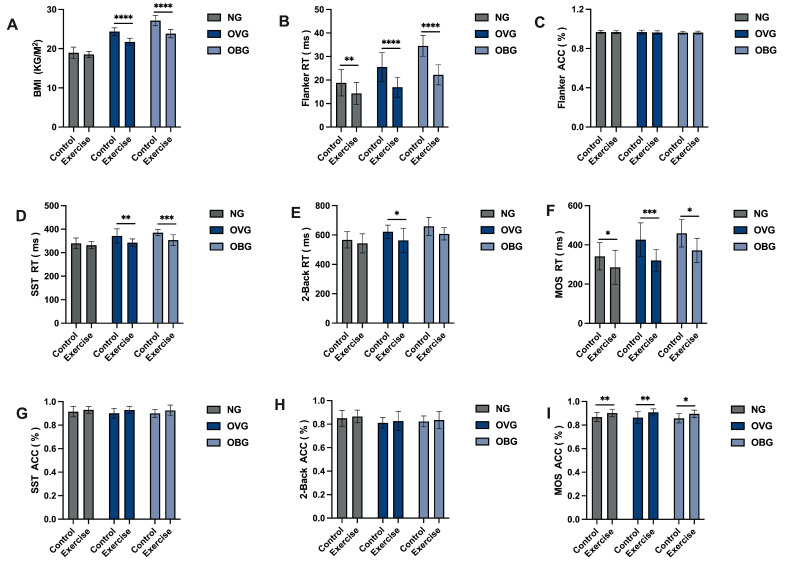
Comparison of EF after exercise intervention. Comparisons of the following parameters after exercise intervention: (**A**) BMI, (**B**) Flanker task reaction (ms), (**C**) Flanker task accuracy (%), (**D**) SST reaction (ms), (**E**) 2-Back reaction (ms), (**F**) MOS task reaction (ms), (**G**) SST accuracy (%), (**H**) 2-Back accuracy (%), and (**I**) MOS task accuracy (%). N-C, Normal weight control group; N-E, normal weight exercise group; OV-C, overweight control group; OV-E, overweight exercise group; OB-C, obese control group; OB-E, obese exercise group; RT, reaction time; ACC, accuracy; * *p* < 0.05, ** *p* < 0.01, *** *p* < 0.001, and **** *p* < 0. 0001.

**Table 1 jfmk-10-00152-t001:** Baseline characteristics of the participants.

Study	Groups	Number	Age	Body Mass Index
		(Female/Male)	(Years Old)	(kg/m^2^)
Part one: The correlation between body mass index and EF	Normal weight group	96 (50/46)	13.11 ± 0.28	19.10 ± 1.52
	Overweight group	47 (23/24)	13.17 ± 0.21	23.43 ± 1.06
	Obese group	23 (9/14)	13.23 ± 0.28	27.10 ± 1.49
Part two: The effect of an 8-week moderate-to-high-intensity rope skipping training in adolescents	Normal weight control group (N-C)	23 (12/11)	13.08 ± 0.31	18.56 ± 1.38
	Normal weight exercise group (N-E)	23 (12/11)	13.15 ± 0.28	18.66 ± 1.37
	Overweight control group (OV-C)	13 (7/6)	13.12 ± 0.15	24.19 ± 0.88
	Overweight exercise (OV-E)	14 (6/8)	13.16 ± 0.28	24.15 ± 0.66
	Obese control group (OB-C)	12 (6/6)	13.24 ± 0.30	27.04 ± 1.05
	Obese exercise group (OB-E)	11 (5/6)	13.19 ± 0.27	26.72 ± 1.05

Note. All values are presented as means ± standard deviation (SD). EF, executive function; N-C: normal weight control group; N-E: normal weight exercise group; OV-C: overweight control group; OV-E: overweight exercise group; OB-C: obese control group; OB-E: obese exercise group.

**Table 2 jfmk-10-00152-t002:** Exercise protocol.

Process	Warm-Up	RS Training	Relaxation
Time	10 min	30 min	10 min
Content	(1) Jogging (2) High knees (3) Butt kicks (4) Bounding (5) Grapevines (6) Slow skipping (7) Lateral bounding (8) Hamstring extension	Content of each set (1) 30 s RS (60–80 jumps) 30 s rest (2) 60 s RS (120–160 jumps) 60 s rest (3) 90 s RS (180–240 jumps) 90 s rest Repeat 5 sets	(1) Thigh stretch (2) Calf stretch (3) Side stretch (4) Lunge stretch
Exercise intensity	50% HRmax–59% HRmax (105–124 beats/minute)	60% HRmax–79% HRmax (125–165 beats/minute)	<50% HRmax (<105 beats/minute)

Note. HRmax: Maximum heart rate; RS: rope skipping. The movements will be uploaded as video attachments.

**Table 3 jfmk-10-00152-t003:** Descriptive statistics and variance analysis of pre-test data.

Groups	NG (*n* = 42)	OVG (*n* = 22)	OBG (*n* = 19)	F	*p*-Value
Flanker RT (ms)	19.24 ± 5.32	26.64 ± 4.54	34.49 ± 4.66	96.617	<0.0001 ****
SST RT (ms)	345.06 ± 22.48	366.80 ± 19.50	386.70 ± 24.06	39.376	<0.0001 ****
2-Back RT (ms)	567.99 ± 50.00	624.27 ± 56.41	670.64 ± 51.02	43.479	<0.0001 ****
MOS RT (ms)	335.93 ± 100.05	428.37 ± 67.73	475.67 ± 91.05	30.387	<0.0001 ****
Flanker ACC (%)	95 ± 2	96 ± 2	96 ± 2	2.375	0.096
SST ACC (%)	90 ± 4	91 ± 3	91 ± 4	2.031	0.135
2-Back ACC (%)	85 ± 6	81 ± 6	81 ± 7	7.906	0.001 ***
MOS ACC (%)	88 ± 4	86 ± 4	85 ± 4	3.581	0.030 *

Note. NG, normal weight group; OVG, overweight group; OBG, obese group; * *p* < 0.05, *** *p* < 0.001, **** *p* < 0.0001.

**Table 4 jfmk-10-00152-t004:** Descriptive statistics and variance analysis of post-test data.

Groups	N-C (*n* = 21)	N-E (*n* = 21)	Bonferroni		OV-C (*n* = 10)	OV-E (*n* = 12)	Bonferroni		OB-C (*n* = 10)	OB-E (*n* = 9)	Bonferroni		ANOVA								
													BMI Grouping			Intervention Way			Main (BMI × Intervention)		
			95% CI	*p*			95% CI	*p*			95% CI	*p*	F	*p*	η_P2_	F	*p*	η_P2_	F	*p*	η_P2_
Flanker RT (ms)	18.83 ± 5.62	14.30 ± 4.68	1.461, 7.595	0.004 **	25.56 ± 6.04	16.92 ± 4.20	4.379, 12.889	0.0001 ****	34.51 ± 4.43	22.21 ± 4.26	7.735, 16.867	0.0001 ****	36.812	0.0001 ****	0.489	53.159	0.0001 ****	0.408	4.189	0.019 *	0.098
SST RT (ms)	339.84 ± 22.87	331.95 ± 15.37	−4.567, 20.358	0.211	371.45 ± 29.80	342.87 ± 15.60	11.294, 45.876	0.002 **	385.36 ± 13.14	353.54 ± 23.02	13.261, 50.370	0.001 ***	20.166	0.0001 ****	0.344	23.160	0.0001 ****	0.231	3.128	0.049 *	0.075
2-Back RT (ms)	566.87 ± 55.49	542.74 ± 65.06	−13.523, 61.784	0.206	622.42 ± 45.50	563.45 ± 82.99	6.726, 111.210	0.027 *	658.72 ± 61.91	607.93 ± 41.73	−5.268, 106.852	0.075	11.125	0.0001 ****	0.224	9.751	0.003 **	0.112	0.686	0.507	0.081
MOS RT (ms)	341.96 ± 69.68	285.99 ± 86.36	10.631, 101.306	0.016 *	426.89 ± 85.24	320.69 ± 55.01	43.294, 169.101	0.001 ***	459.10 ± 69.75	372.29 ± 61.57	19.311, 154.312	0.012 *	13.584	0.0001 ****	0.261	23.257	0.0001 ****	0.232	0.895	0.413	0.023
Flanker ACC (%)	96 ± 1	96 ± 1	−0.8, 1.2	0.659	96 ± 2	96 ± 1	−1.0, 1.7	0.596	96 ± 1	96 ± 1	−1.6, 1.3	0.869	1.095	0.340	0.028	0.152	0.697	0.002	0.120	0.887	0.003
SST ACC (%)	91 ± 4	93 ± 2	−3.8, 0.8	0.189	90 ± 4	92 ± 2	−6.0, 0.4	0.082	90 ±3	92 ±4	−6.0, 0.9	0.140	0.540	0.585	0.014	6.980	0.010 *	0.083	0.261	0.771	0.007
2-Back ACC (%)	85 ± 6	86 ± 5	−5.4, 2.3	0.432	81 ± 4	82 ± 8	−6.9, 3.8	0.560	82 ± 4	83 ± 7	−6.9, 4.6	0.691	3.221	0.045*	0.077	0.937	0.336	0.012	0.007	0.993	0.000
MOS ACC (%)	83 ± 3	90 ± 3	−5.7, −1.2	0.003 **	86 ± 4	90 ± 2	−7.5, −1.2	0.007 **	85 ± 3	89 ± 3	−7.5, −5	0.025 *	0.415	0.661	0.011	20.703	0.0001 ****	0.212	0.103	0.903	0.003

Note. All values are presented as means ± standard deviation (SD). N-C, normal weight control group; N-E, normal weight exercise group; OV-C, overweight control group; OV-E, overweight exercise group; OB-C, obese control group; OB-E, obese exercise group; RT, reaction time; ACC, accuracy; ms, millisecond; * *p* < 0.05, ** *p* < 0.01, *** *p* < 0.001, **** *p* < 0.0001.

## Data Availability

The data and materials this study is based on are available from the corresponding author, L.Z.

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
