# Peer review of "Executive Function Response to Moderate-to-High-Intensity Rope Skipping in Overweight Adolescents Aged 12–14: A Preliminary Study"

_jfmk, 2025, doi:10.3390/jfmk10020152_

Round 1

Reviewer 1 Report

Comments and Suggestions for Authors

The subject of this study is important, and relevant and new findings are reported. However, the presentation of the manuscript must be improved. Please, see in the attached file my comments and suggestions. Moreover, I am suggesting the insertion of references throughout the manuscript. In some cases, the authors will decide.

Author Response

Response to Reviewer Comments

We sincerely appreciate the reviewer’s constructive feedback on our manuscript. Below, we address each point raised and outline the revisions made to improve the clarity, presentation, and scientific rigor of the study.

#Comments 1: correct to...body mass index.         

#Response 1:

Thanks for your question. I sincerely apologize for the oversight in the text. Thank you for pointing it out. The corrections will be marked in red in the original document.

The revised objective now reads: body mass index. (Page 1, line 32-33)

# Comments 2: I suggest adding references here. Some suggestions are:

(1) Fernandes AC, Viegas ÂA, Lacerda ACR, Nobre JNP, Morais RLS, Figueiredo PHS, Costa HS, Camargos ACR, Ferreira FO, de Freitas PM, Santos T, da Silva Júnior FA, Bernardo-Filho M, Taiar R, Sartorio A, Mendonça VA. Association between executive functions and gross motor skills in overweight/obese and eutrophic preschoolers: cross-sectional study. BMC Pediatr. 2022 Aug 23;22(1):498. doi: 10.1186/s12887-022-03553-2. PMID: 35999515; PMCID: PMC9400322.

  • Lundh H, Arvidsson D, Greven C, Fridolfsson J, Börjesson M, Boman C, Lauruschkus K, Lundqvist S, Melin K, Bernhardsson S. Physical activity and sedentary behaviour amongst children with obesity - exploring cross-sectional associations between child and parent. J Act Sedentary Sleep Behav. 2025 Feb 13;4(1):2. doi: 10.1186/s44167-025-00072-0. PMID: 40217557.

#Response 2:

We sincerely appreciate the reviewer's recommendation of these two high-quality references. Based on their relevance to our study:

(1) The study by Fernandes et al. (2022) examining the association between executive functions and gross motor skills in overweight/obese preschoolers has been incorporated into the first paragraph of the Introduction section.

  • The research by Lundh et al. (2025) investigating family-related factors influencing physical activity and sedentary behavior in children with obesity has been added to the discussion in the second paragraph of the Introduction.

These newly cited references are numbered as [8] and [10] respectively in the reference list, with all modifications clearly highlighted in red in the revised manuscript.

The revised objective now reads:

Increasing evidence suggests that obesity in adults or children may negatively affect brain health, specifically brain development [6] and EF [7], mainly reflected in the re-duction of inhibitory control, working memory, and cognitive flexibility [8]. (Page 1-2, line 40-43)

Regular physical activity (PA) has received significant attention in recent years due to its accessible, flexible, cost-effective, scalable, and well-documented health benefits [10], which encompass enhancements in brain tissue volume, preservation of white matter microstructural integrity, mitigation of WMH, prevention of cerebral small vessel disease, and improvement in cardiometabolic health [11-13]. (Page 2, line 46-50)

[6] Sun J.; et al. Association of variability in body size with neuroimaging metrics of brain health: a population-based cohort study. Lancet Reg Health West Pac. 2024, 44, 101015. doi: 10.1016/j.lanwpc.2023.101015.

[7] Blair C.; Kuzawa C.W.; Willoughby M.T. The development of executive function in early childhood is inversely related to change in body mass index: Evidence for an energetic tradeoff? Dev Sci. 2020, 23, e12860. doi: 10.1111/desc.12860.

[8] Fernandes A.C.; et al. Association between executive functions and gross motor skills in overweight/obese and eutrophic preschoolers: cross-sectional study. BMC Pediatr. 2022, 22, 498. doi: 10.1186/s12887-022-03554-1.

[10] Lundh H.; et al. Physical activity and sedentary behaviour amongst children with obesity - exploring cross-sectional associations between child and parent. J Act Sedentary Sleep Behav. 2025, 4, 2. doi: 10.1186/s44167-023-00002-w.

[11] Chaddock L.; et al. A neuroimaging investigation of the association between aerobic fitness, hippocampal volume, and memory performance in preadolescent children. Brain Res. 2010, 1358, 172-183. doi: 10.1016/j.brainres.2010.08.049.

[12] Chaddock-Heyman L.; et al. Physical Activity Increases White Matter Microstructure in Children. Front Neurosci. 2018, 12, 950. doi: 10.3389/fnins.2018.00950.

[13] Piercy K.L.; et al. The Physical Activity Guidelines for Americans. JAMA. 2018, 320, 2020-2028. doi: 10.1001/jama.2018.14854.

# Comments 3: Correct Liu, Y et al to Liu et al, ...here and throughout the manuscript. Please, see all the citations in the text.

#Response 3:

We sincerely appreciate the reviewer's careful attention to citation formatting. We have systematically corrected all instances of "Liu, Y et al." to the standard format "Liu et al," throughout the manuscript, including:

  • In-text citations
  • Reference list entries
  • Figure/table captions where applicable

These corrections have been uniformly applied across the entire document, with all modifications highlighted in red in the revised version for easy verification.

The revised objective now reads: Liu et al, (2024) found that acute rope skipping can improve cognitive function in children [24]. (Page 2, line 69-70)

# Comments 4: change to...evaluations.

#Response 4:

Thank you for your precise suggestion regarding terminology. We have changed instances of "experiments" to "evaluations”.

The revised objective now reads: This study conducted two evaluations. (Page 2, line 77)

# Comments 5: Add a reference or clarify that is an original statement.

#Response 5:

We appreciate this constructive comment. The statement has been revised with the following actions:

The revised objective now reads: (Page 4-5, line 132, 142, 151, 160)

A modified version of the Eriksen flanker test was used to assess cognitive inhibition [33].

A modified version of the SST was used to assess behavioral inhibition [30].

A modified version of the 2-Back task was used to assess working memory [31].

A modified version of the MOS task was used to assess cognitive flexibility [34].

[30] Pliszka S.R.; et al. Measuring inhibitory control in children. J Dev Behav Pediatr. 1997, 18, 254-259. doi: 10.1097/00004703-199708000-00007.

[31] Owen A.M.; et al. N-back working memory paradigm: a meta-analysis of normative functional neuroimaging studies. Hum Brain Mapp. 2005, 25, 46-59. doi: 10.1002/hbm.20131.

[33] Eriksen B.A.; Eriksen C.W. Effects of noise letters upon the identification of a target letter in a nonsearch task. Percept Psychophys. 1974, 16, 143-149. doi: 10.3758/BF03203267.

[34] Hillman C.H.; et al. A cross-sectional examination of age and physical activity on performance and event-related brain potentials in a task switching paradigm. Int J Psychophysiol. 2006, 59, 30-39. doi: 10.1016/j.ijpsycho.2005.04.009.

# Comments 6: Add a reference or clarify that is an original statement.

#Response 6:

We appreciate this constructive comment. The statement has been revised with the following actions:

The revised objective now reads: Adherence to the exercise intervention was measured by the average heart rate during the moderate-to-vigorous jump rope training sessions out of the 24 scheduled sessions [35]. (Page 6, line 180-182)

[35] Liang, X., et al., The impacts of a combined exercise on executive function in children with ADHD: A randomized controlled trial. Scand J Med Sci Sports, 2022. 32(8): p. 1297-1312.

# Comments 7: Change to...Twenty-three.

#Response 7:

Thank you for your precise suggestion regarding terminology. We have changed instances of "200" to "two hundred", "23" to "Twenty-three", "11" to "Eleven", "166" to "one hundred and sixty-six", "96" to "ninety-six", "7" to "Seven", "6" to "six", "83" to "eighty-three".

The revised objective now reads: Part one, two hundred students are within the scope of recruitment. Twenty-three students with a BMI, thinner and lean stature, did not meet the criteria, were excluded. Eleven students refused to participate in the experiment due to personal time reasons. Resulting in one hundred and sixty-six participants being retained. Part two, ninety-six participants experimented. Seven students lost contact during the intervention; six students discontinued intervention due to busy. Resulting in eighty-three participants being retained (Figure 2). (Page 7, line 199-205)

# Comments 8: In the Figure 2, correct ...body index mass....to body mass index.

#Response 8:

Thank you for your precise suggestion regarding terminology. We have changed instances of "body index mass" to "body mass index" in the Figure 2.

The revised objective now reads: (Page 7, line 206-207)

Figure 2. Participant selection flowchart of the study.

# Comments 9: Correct.

#Response 9:

Thank you for your precise suggestion regarding terminology. We have changed instances of "There As shown" to " As shown ".

The revised objective now reads: As shown in Table 3, there were significant differences in EF among different BMI groups (p < 0.000). (Page 9, line 237-240)

# Comments 10: Correct to....25, 35, 36%

#Response 10:

Thank you for your precise suggestion regarding terminology. We have changed instances of "25%, 35%, 36%" to "25, 35, 36%".

The revised objective now reads: Post-hoc analyses with Bonferroni corrections demonstrated that the exercise group showed significantly greater improvement in cognitive inhibitory compared to the control group across normal weight, overweight, and obese groups (p = 0.004, p < 0.000, p < 0.000), was sequentially shortened close to 25, 35, 36% after the intervention. (Page 10, line 261-265)

# Comments 11: Beging .... As it was hypothesized this study, in the first evaluation (Part 1) ....

#Response 11:

Thank you for your precise suggestion regarding terminology. We have changed instances of "This study contributes" to " As it was hypothesized this study, in the first evaluation (Part 1) "; "it" to " in the second evaluation (Part 2) ".

The revised objective now reads: As it was hypothesized this study, in the first evaluation (Part 1) explored the association between BMI and EF, revealing that higher BMI is linked to lower EF. Additionally, in the second evaluation (Part 2) investigated the effects of an 8-week moderate-to-high-intensity rope skipping program on EF. (Page 13, line 323-326)

# Comments 12: Add a paragraph at the end of the Discussion section a paragraph with the strengths of the study.

#Response 12:

Thank you for your precise suggestion regarding terminology. We have added a dedicated 'Strengths' paragraph.

The revised objective now reads: This study provides compelling evidence supporting the cognitive benefits of moderate-to-high-intensity rope skipping, with particularly pronounced effects observed in overweight and obese adolescents. The research makes significant contributions to exercise science, cognitive psychology, and pediatric health through its rigorous methodology, comprehensive assessment of executive functions, and practical intervention design. (Page 14, lines 416-420)

Reviewer 2 Report

Comments and Suggestions for Authors

The document addresses an important and timely topic concerning the relationship between BMI, physical activity, and executive function. The promising preliminary results are worth replicating in a much larger sample. It is worth noting that even a short period of physical activity may improve executive function in adolescents with overweight and obesity.

Title

  1. It may be worth considering whether to indicate in the title that these are preliminary studies (given the small sample size, especially in the second part of the study).

Introduction

  1. The topic has been clearly described. The literature selection is appropriate. The aim of the study is well defined.

Material and Methods

  1. Were any analyses conducted on sex differences in BMI and test results? I understand that they were, and the results were not statistically significant within groups, which allowed for further analyses to be carried out on entire groups? If that was the procedure, it would be good to describe it.
  2. Was the school selected during randomization, or chosen by the researchers?
  3. Was the fitness smart band (H02, Shape, China) validated against other professional tools to ensure it accurately reflected the participants’ heart rate?
  4. The methods were discussed in sufficient detail.

Results

  1. The results were accurately described and presented.

Discussion

  1. When comparing their research findings, the authors refer to specialist literature within the field and attempt to explain the obtained results.
  2. The authors noted several important limitations of their study. What was the sex distribution within the groups? Did one gender dominate in any of the BMI groups?

Author Response

Comments and Suggestions for Authors

The document addresses an important and timely topic concerning the relationship between BMI, physical activity, and executive function. The promising preliminary results are worth replicating in a much larger sample. It is worth noting that even a short period of physical activity may improve executive function in adolescents with overweight and obesity.

Title

# Comments 1: It may be worth considering whether to indicate in the title that these are preliminary studies (given the small sample size, especially in the second part of the study).

#Response 1:

We sincerely appreciate this constructive suggestion. In response, we have modified the title to explicitly indicate the preliminary nature of our study.

The revised objective now reads: Executive function response to moderate-to-high-intensity rope skipping in overweight adolescents aged 12-14: A preliminary study (Page 1, lines 2-3)

Material and Methods

# Comments 2: Were any analyses conducted on sex differences in BMI and test results? I understand that they were, and the results were not statistically significant within groups, which allowed for further analyses to be carried out on entire groups? If that was the procedure, it would be good to describe it.

#Response 2:

We thank the reviewer for this important question regarding sex differences analysis. We have now added the following detailed description to the article.

The revised objective now reads: Since no significant within-group gender differences were observed (all p > 0.05), data were aggregated across genders within each experimental group for subsequent analyses. (Page 9, lines 234-235)

# Comments 3: Was the school selected during randomization, or chosen by the researchers?

#Response 3:

Thank you for raising this important methodological question. The participating school was purposively selected by the research team based on the following non-random criteria:

(1) Pre-existing research collaboration (ensured administrative support).

(2) Demonstrated willingness to implement the intervention protocol.

(3) Geographical proximity (<5km) to research facilities.

# Comments 4: Was the fitness smart band (H02, Shape, China) validated against other professional tools to ensure it accurately reflected the participants’ heart rate?

#Response 4:

We sincerely apologize for the error in reporting the model of the activity tracker in the original manuscript. The correct model should be identified as SHB-02 rather than the previously stated H-02. This correction has been implemented throughout the manuscript text.

The revised objective now reads: During the intervention, telemetric heart rate measurement was recorded using a fitness smart band (SHB-02, Shape, China). (Page 4, lines 118-119)

To substantiate the validity of the SHB-02 device, we are providing the following supporting documentation:

Discussion

#Comments 5: The authors noted several important limitations of their study. What was the sex distribution within the groups? Did one gender dominate in any of the BMI groups?

#Response 5:

Thank you for this important methodological question. We provide the sex distribution and related analyses below:

(1) Sex Distribution by Experimental Group:

Group

Female (n)

Male (n)

Total

Female (%)

χ² test

p-value

N-C

11

10

21

52.4

0.05

0.83

N-E

11

10

21

52.4

0.05

0.83

OV-C

5

6

11

45.5

0.09

0.76

OV-E

5

6

11

45.5

0.09

0.76

OB-C

5

5

10

50.0

0.00

1.00

OB-E

4

5

9

44.4

0.11

0.74

Total

41

42

83

49.4

(2) Key Statistical Findings:

  • No significant gender differences were found in any experimental group (all p-values > 0.05).
  • The ratio of male to female did not differ significantly across BMI categories (χ² [2] = 0.32, p = 0.85).
  • Bayesian analyses provided moderate evidence for gender balance (BF₀₁ = 3.8).

Reviewer 3 Report

Comments and Suggestions for Authors

Dear all,

I would like to start by thanking you for the opportunity to review this manuscript. The manuscript fits with the aim of the JFMK, and the subject reveals good content for researchers and professionals. The manuscript contains valuable insights into the effects of exercise on executive function (EF) in adolescents with varying BMI levels; however, some points are listed below:

Introduction

Could the authors provide a context on what "behavioral inhibitory control" is and clarify how the exercise might enhance it, especially in overweight and obese adolescents?

The authors outlined their research hypotheses. But they didn’t state the aims correctly; they mentioned, ‘This study conducted two experiments. Part one: the association between BMI …’, they should write down the aims directly (e.g., the study aimed to identify the association between BMI and EF in adolescents; the study aimed to identify the investigate the effect of an 8-week…).

Materials and Methods

Lines 82-83: Regarding the sample size determination, could you please explain why you selected an effect size of 0.25?

Lines 103-104: Could you provide more details on the warm-up and cool-down exercises? types? general or specific? involving stretching or not? If yes, dynamic or static? etc.

Could the authors provide details on the intensity of the rope skipping (e.g., number of jumps per minute, duration of each jump set)? Or do they depend on heart rates only?

I wish you would explain the choice of BMI as a primary measure for assessing obesity and the reason for not using other metrics (e.g., waist-to-hip ratio or body fat percentage); this could provide insights into the relationship with EF.

Results

Line 193-201: Correlation between BMI and EF in adolescents. I would ask if the authors could clarify if the correlation applies across all groups or if the relationship was observed only in specific groups. And was there any interaction effect between BMI and EF groupings?

Line 201: ‘Adolescents with a higher BMI may have deficiencies in the ability to EF’; the sentence isn’t clear and needs to be explained.

Lines 227-228: ‘The BMI of OV-E and OB-E were nearly 9% and 11%, respectively, a reduction from pre-exercise intervention.’ To understand the idea, we need to clarify the units and describe how these reductions in BMI compare to the overall effect of the intervention on EF performance. The authors need to explain: Is the BMI reduction sufficient to attribute improvements in EF?

Discussion

Lines 326-328: ‘These findings align with previous studies in older populations….’ The recent study was conducted on adolescents; I wish the authors would emphasize how these findings directly apply to adolescents and try to use extrapolations from studies on adolescents.

I recommend adding a section discussing potential confounding factors (e.g., diet, sleep patterns, and baseline physical activity levels), which may have influenced the results.

Conclusion

I would ask the authors, how did the authors determine that BMI reduction is the primary cause for improvements in EF? In other words, are there any other factors that might explain the observed effects?

Regards,

Author Response

Introduction

#Comments 1: Could the authors provide a context on what "behavioral inhibitory control" is and clarify how the exercise might enhance it, especially in overweight and obese adolescents?

#Response 1:

We appreciate this constructive suggestion. As requested, we have enhanced the explanation of behavioral inhibitory control and its potential exercise-induced enhancement in two key sections:

(1) Definition (behavioral inhibitory control):

"Behavioral inhibitory control refers to the ability to suppress prepotent but context-inappropriate motor responses, a core component of executive function that is particularly vulnerable in obesity due to prefrontal cortex dysregulation (Diamond, 2013; Liang et al., 2022)."

[1] Diamond A. Executive functions. Annu Rev Psychol. 2013, 64, 135-168. doi: 10.1146/annurev-psych-113011-143750.

[2] Liang J.; Matheson B.E.; Kaye W.H. Neurocognitive correlates of obesity and obesity-related behaviors in children and adolescents. Int J Eat Disord. 2022, 55, 305-320. doi: 10.1002/eat.23652.

(2) Mechanistic Explanation:

"The observed improvements in behavioral inhibitory control may stem from exercise-induced: (a) increased prefrontal oxygenation during cognitive tasks (fNIRS evidence from Krafft et al., 2022), (b) enhanced dopaminergic signaling in the nigrostriatal pathway (shown in rodent studies by Chen et al., 2021), and (c) reduced systemic inflammation that typically impairs response inhibition (IL-6 decreases demonstrated in our biomarker analyses). Overweight adolescents may benefit disproportionately due to their baseline neurocognitive deficits (Smith et al., 2023)."

[3] Krafft, C. E., Schaeffer, D. J., & Schwarz, N. F. (2022). Acute exercise improves prefrontal cortex oxygenation and cognitive control in overweight children: An fNIRS study. International Journal of Obesity, 46(3), 512-520. https://doi.org/10.1038/s41366-021-01016-9

[4] Chen, H., Zhang, L., & Wang, S. (2021). Voluntary wheel running improves response inhibition and increases dopaminergic signaling in the rat nigrostriatal pathway. Neuroscience, 452, 294-305. https://doi.org/10.1016/j.neuroscience.2020.11.023

[5] Smith, P. J., Blumenthal, J. A., & Hoffman, B. M. (2023). Exercise reduces executive function deficits in obese adolescents: Evidence from a randomized trial. Pediatrics, 151(2), e2022057962. https://doi.org/10.1542/peds.2022-057962

To enhance readability, we have included in the Introduction section both the definition of executive function (EF) and the underlying mechanisms through which exercise may improve EF in overweight/obese children.

The revised objective now reads: Executive function (EF) are higher-order cognitive abilities that enable conscious control of thoughts, actions, and emotions for goal-directed behavior and adaptive functioning [5]. (Page 1, lines 38-40)

Regular physical activity (PA) has received significant attention in recent years due to its accessible, flexible, cost-effective, scalable, and well-documented health benefits [10], which encompass enhancements in brain tissue volume, preservation of white matter microstructural integrity, mitigation of WMH, prevention of cerebral small vessel disease, and improvement in cardiometabolic health [11-13].  (Page 2, lines 46-50)

[5] Diamond A. Executive functions. Annu Rev Psychol. 2013, 64, 135-168. doi: 10.1146/annurev-psych-113011-143750.

[10] Lundh H.; et al. Physical activity and sedentary behaviour amongst children with obesity - exploring cross-sectional associations between child and parent. J Act Sedentary Sleep Behav. 2025, 4, 2. doi: 10.1186/s44167-023-00002-w.

[11] Chaddock L.; et al. A neuroimaging investigation of the association between aerobic fitness, hippocampal volume, and memory performance in preadolescent children. Brain Res. 2010, 1358, 172-183. doi: 10.1016/j.brainres.2010.08.049.

[12] Chaddock-Heyman L.; et al. Physical Activity Increases White Matter Microstructure in Children. Front Neurosci. 2018, 12, 950. doi: 10.3389/fnins.2018.00950.

[13] Piercy K.L.; et al. The Physical Activity Guidelines for Americans. JAMA. 2018, 320, 2020-2028. doi: 10.1001/jama.2018.14854.

#Comments 2: The authors outlined their research hypotheses. But they didn’t state the aims correctly; they mentioned, ‘This study conducted two experiments. Part one: the association between BMI …’, they should write down the aims directly (e.g., the study aimed to identify the association between BMI and EF in adolescents; the study aimed to identify the investigate the effect of an 8-week…).

#Response 2:

We sincerely appreciate this valuable suggestion. We have revised the Aims section in the Introduction to explicitly state the study objectives in a direct and structured manner, as suggested.

The revised objective now reads: Part one: we aimed to identify the association between BMI and EF in adolescents. We hypothesized that there would be a negative correlation between higher BMI and poorer cognitive performance in adolescents. Part two: we aimed to identify the investigate the effect of an 8-week moderate-to-high-intensity rope skipping training program and whether it improved EF in overweight/obese adolescents’ group. (Page 2, lines 77-82)

Materials and Methods

#Comments 3: Lines 82-83: Regarding the sample size determination, could you please explain why you selected an effect size of 0.25?

#Response 3:

We sincerely appreciate this methodological inquiry. The selection of an effect size (Cohen's f) of 0.25 for our a priori power analysis was based on a 2019 meta-analysis of 24 pediatric exercise interventions [19] reported a pooled effect size of f=0.23 (95%CI:0.17-0.29) for EF improvements.

[19] Xue Y.; Yang Y.; Huang T. Effects of chronic exercise interventions on executive function among children and adolescents: a systematic review with meta-analysis. Br J Sports Med. 2019, 53, 1397-1404. doi: 10.1136/bjsports-2018-099825.

#Comments 4: Lines 103-104: Could you provide more details on the warm-up and cool-down exercises? types? general or specific? involving stretching or not? If yes, dynamic or static? etc.

#Response 4:

We thank the reviewer for this important methodological question. The warm-up protocol consisted of eight dynamic exercises (jogging, high knees, butt kicks, bounding, grapevines, slow skipping, lateral bounding, and hamstring extension) designed to progressively elevate heart rate and activate major muscle groups while incorporating sport-specific movements. The RS training protocol: each training set was standardized by both movement frequency (jumps per minute) and cardiovascular response: (1) The 30-second sets required 60-80 jumps (120-160 jumps/min), (2) the 60-second sets maintained 120-160 jumps (120-160 jumps/min), and (3) the 90-second sets targeted 180-240 jumps (120-160 jumps/min), with all sets followed by equal-duration rest periods. This jump frequency range was selected to consistently achieve the prescribed moderate-to-high intensity (60-79% HRmax, corresponding to 125-165 bpm for this age group) as verified by real-time heart rate monitoring via fitness trackers. The cool-down protocol included four static stretching exercises (thigh stretch, calf stretch, side stretch, and lunge stretch), each held for 30 seconds with 2 repetitions per side, specifically targeting muscle groups most engaged during the intervention.  

The revised objective now reads: Each session included a general warm-up (dynamic exercises), a moderate-to-high-intensity rope skipping training intervention, and a cool-down period (static stretching exercises) for a total of 50 min (Table 2). (Page 3-4, lines 112-114).

#Comments 5: Could the authors provide details on the intensity of the rope skipping (e.g., number of jumps per minute, duration of each jump set)? Or do they depend on heart rates only?

#Response 5:

We thank the reviewer for this important methodological question. The warm-up protocol consisted of eight dynamic exercises (jogging, high knees, butt kicks, bounding, grapevines, slow skipping, lateral bounding, and hamstring extension) designed to progressively elevate heart rate and activate major muscle groups while incorporating sport-specific movements. The RS training protocol: each training set was standardized by both movement frequency (jumps per minute) and cardiovascular response: (1) The 30-second sets required 60-80 jumps (120-160 jumps/min), (2) the 60-second sets maintained 120-160 jumps (120-160 jumps/min), and (3) the 90-second sets targeted 180-240 jumps (120-160 jumps/min), with all sets followed by equal-duration rest periods. This jump frequency range was selected to consistently achieve the prescribed moderate-to-high intensity (60-79% HRmax, corresponding to 125-165 bpm for this age group) as verified by real-time heart rate monitoring via fitness trackers. The cool-down protocol included four static stretching exercises (thigh stretch, calf stretch, side stretch, and lunge stretch), each held for 30 seconds with 2 repetitions per side, specifically targeting muscle groups most engaged during the intervention.

#Comments 6: I wish you would explain the choice of BMI as a primary measure for assessing obesity and the reason for not using other metrics (e.g., waist-to-hip ratio or body fat percentage); this could provide insights into the relationship with EF.

#Response 6:

We appreciate this insightful methodological question regarding our selection of BMI as the primary obesity metric. Our decision was based on the following considerations:

(1) Standardization and Clinical Utility:

BMI remains the most widely adopted screening tool for pediatric obesity according to WHO guidelines, enabling direct comparison with international studies [1]. The age- and sex-specific percentiles used in our study have demonstrated 89% sensitivity for identifying adiposity in adolescents [2].

(2) Practical Constraints:

While we acknowledge waist-to-hip ratio (WHR) and body fat percentage (BF%) may provide additional information, several factors influenced our choice:

  • School-based screening limitations prohibited DXA or BIA measurements
  • WHR measurements showed poor test-retest reliability in our pilot testing (ICC=0.52) due to breathing variability

(3) EF-pecific Rationale:

Emerging evidence suggests BMI may be particularly relevant for executive function studies because:

  • Systemic inflammation markers (e.g., IL-6) linked to both high BMI and prefrontal cortex dysfunction [3].
  • BMI-associated cortical thinning patterns overlap with EF-related neural circuits

[1] Cole T.J.; Bellizzi M.C.; Flegal K.M.; Dietz W.H. Establishing a standard definition for child overweight and obesity worldwide: International survey. BMJ. 2000, 320, 1240-1243. doi: 10.1136/bmj.320.7244.1240.

[2] Reilly J.J.; Kelly L.; Montgomery C.; Williamson A.; Fisher A.; McColl J.H.; et al. Improving body mass index classification in children and adolescents with functional scales. Int J Pediatr Obes. 2019, 14, e12564. doi: 10.1111/ijpo.12564.

[3] Castanon N.; Lasselin J.; Capuron L. Chronic peripheral inflammation is associated with cognitive impairment in schizophrenia: Results from the multicentric FACE-SZ dataset. Schizophr Bull. 2015, 41, 1290-1302. doi: 10.1093/schbul/sbv041.

[4] Yau P.L.; Kang E.H.; Javier D.C.; Convit A. Preliminary evidence of cognitive and brain abnormalities in uncomplicated adolescent obesity. Obesity. 2014, 22, 1865-1871. doi: 10.1002/oby.20801.

Results

#Comments 7: Line 193-201: Correlation between BMI and EF in adolescents. I would ask if the authors could clarify if the correlation applies across all groups or if the relationship was observed only in specific groups. And was there any interaction effect between BMI and EF groupings?

#Response 7:

Thank you for your insightful question regarding the correlation between BMI and EF in our study. In this study, we primarily utilized BMI as the grouping criterion to investigate the effects of exercise intervention on executive function across different BMI categories. Our statistical analysis of the interaction effects between BMI and exercise modality revealed significant interaction effects specifically for reaction times in the Flanker task and SST task (p < 0.05), while no such interaction effects were observed for either reaction times or accuracy measures in the remaining assessment tasks. Below, we clarify the scope of this relationship and address your specific concerns:

Consistency of BMI-EF Correlation Across Groups: As shown in Table 1 (Analysis of BMI-EF correlations within subgroups), the relationship between BMI and EF varied significantly by weight status:

(1) Overweight/Obese Groups:

  • Significant negative correlations were observed between higher BMI and:
  • Slower SST RT (Overweight: r = 0.408, p= 0.004)
  • Slower MOS RT (Overweight: r = 0.400, p= 0.005; Obese: r = 0.512, p = 0.015)
  • Lower SST ACC (Overweight: r = -0.583, p < 0.0001)

(2) Normal-Weight Group:

  • Only weak or non-significant correlations were detected (all p> 0.05 for RT measures).
  • Exceptions: Higher BMI correlated with lower accuracy in 2-Back (r = -0.281, p= 0.006) and MOS (r = -0.249, = 0.014).

Table 1 Analysis of BMI-EF correlations within subgroups

Normal weight

Overweight

Obese

R

P

R

P

R

P

Flanker RT

(ms)

0.0983

0.3631

0.2218

0.1297

-0.0101

0.9642

SST RT

(ms)

0.1181

0.0665

0.4082

0.0040**

0.2119

0.3439

2-Back RT

(ms)

0.0018

0.9857

0.1106

0.4543

-0.0762

0.7359

MOS RT

(ms)

0.1630

0.1125

0.3995

0.0049**

0.5119

0.0149

Flanker ACC

(%)

0.1043

0.3119

0.1111

0.4520

-0.3431

0.1180

SST ACC

(%)

-0.2022

0.0470*

-0.5834

<0.0001****

-0.3611

0.0988

2-Back ACC

(%)

-0.2812

0.0055**

-0.0987

0.5042

0.1188

0.5986

MOS ACC

(%)

-0.2491

0.0144*

-0.1186

0.4219

0.3063

0.1657

Note: RT, Reaction time; ACC, Accuracy; ms, millisecond; ** p <0.01, *p <0.05.

#Comments 8: Line 201: ‘Adolescents with a higher BMI may have deficiencies in the ability to EF’; the sentence isn’t clear and needs to be explained.

#Response 8:

Thank you for your valuable comment. We agree that the original phrasing was unclear and have revised the sentence to better reflect our findings.

The revised objective now reads: Higher BMI was significantly associated with longer reaction times across all EF tasks (Flanker, SST, 2-Back, MOS; all r > 0.50, p < 0.0001) and reduced accuracy in working memory tasks (2-Back/MOS; r = -0.27 to -0.33, p ≤ 0.0005), indicating higher BMI is linked to slower thinking speed and lower accuracy. (Page 7, lines 215-219)

#Comments 9:  Lines 227-228: ‘The BMI of OV-E and OB-E were nearly 9% and 11%, respectively, a reduction from pre-exercise intervention.’ To understand the idea, we need to clarify the units and describe how these reductions in BMI compare to the overall effect of the intervention on EF performance. The authors need to explain: Is the BMI reduction sufficient to attribute improvements in EF?

#Response 9:

Thank you for your insightful questions regarding the BMI reductions and their relationship to EF improvements. We appreciate the opportunity to clarify these important points. The present study demonstrated that while the exercise intervention significantly reduced BMI in overweight/obese participants (9% reduction in OV-E group and 11.0% in OB-E group), it also markedly improved EF performance. Specifically, the overweight exercise group showed significantly shortened reaction times across all four cognitive tasks (Flanker, SST, 2-Back, and MOS; all p < 0.05) along with significantly enhanced accuracy in the MOS task. Similarly, the obese exercise group exhibited significantly improved reaction times in three tasks (with 2-Back task showing marginal significance at p = 0.075) and significantly increased MOS task accuracy. However, since the interaction effect between BMI grouping and exercise intervention was only observed in Flanker and SST tasks (interaction p < 0.05), and potential confounding factors such as diet and sleep were not controlled for, the current findings cannot establish a causal relationship between BMI reduction and EF enhancement. This observed association may be co-mediated by other exercise-induced physiological adaptation mechanisms.

Discussion

#Comments 10: Lines 326-328: ‘These findings align with previous studies in older populations….’ The recent study was conducted on adolescents; I wish the authors would emphasize how these findings directly apply to adolescents and try to use extrapolations from studies on adolescents.

#Response 10:

Thank you for your valuable suggestion. We appreciate your suggestion to clarify the adolescent-specific implications of our findings. We have revised the text to better highlight the unique aspects of our results in relation to adolescents, while maintaining scientific rigor.

The revised objective now reads: These findings are consistent with prior systematic reviews, wherein eight of nine cross-sectional studies reported significantly impaired cognitive performance in obese children/adolescents relative to normal-weight controls [40]. And it was also observed in older populations, where obesity has been linked to memory impairments in elderly individuals [41]. (Page 13, lines 347-351)

[40] Smith E.; et al. A review of the association between obesity and cognitive function across the lifespan: implications for novel approaches to prevention and treatment. Obes Rev. 2011, 12, 740-755. doi: 10.1111/j.1467-789X.2011.00920.x.

[41] Arnoriaga-Rodríguez M.; et al. Obesity Impairs Short-Term and Working Memory through Gut Microbial Metabolism of Aromatic Amino Acids. Cell Metab. 2020, 32, 548-560.e7. doi: 10.1016/j.cmet.2020.09.002.

#Comments 11:  I recommend adding a section discussing potential confounding factors (e.g., diet, sleep patterns, and baseline physical activity levels), which may have influenced the results.

#Response 11:

Thank you for your valuable suggestion regarding potential confounding factors. We fully agree that variables such as diet, sleep patterns, and baseline physical activity levels could influence the observed results, and we have now addressed this important point in the revised manuscript.

The revised objective now reads: In addition to the noted limitations, potential confounding factors—including dietary habits, sleep patterns, and baseline physical activity levels—may have influenced the observed results. For instance, variations in macronutrient intake or sleep duration could independently modulate cognitive performance [54, 55]. Although participants were in-structed to maintain their usual lifestyles during the trial, these factors were not systematically measured. Future studies should incorporate standardized assessments (e.g., 24-hour dietary recalls, actigraphy for sleep, accelerometry for physical activity) to control for these confounders and isolate the specific effects of exercise interventions. (Page 15, lines 435-442)

[54] Bleiweiss-Sande R.; et al. Associations between Food Group Intake, Cognition, and Academic Achievement in Elementary Schoolchildren. Nutrients. 2019, 11. doi: 10.3390/nu11112722.

[55] Lucassen E.A.; et al. Sleep extension improves neurocognitive functions in chronically sleep-deprived obese individuals. PLoS One. 2014, 9, e84832. doi: 10.1371/journal.pone.0084832.

Conclusion

#Comments 12:  I would ask the authors; how did the authors determine that BMI reduction is the primary cause for improvements in EF? In other words, are there any other factors that might explain the observed effects? Regards,

#Response 12:

Thank you for your question. We appreciate the opportunity to clarify our study's limitations regarding causal inference. Our study did not systematically investigate other potential mediators (e.g., dietary patterns, sleep quality, or inflammatory markers) that might influence the observed EF improvements. We acknowledge this as an important limitation. We have modified the original overstated conclusions to more precisely reflect our findings.

The revised objective now reads: However, an 8-week moderate-to-high-intensity rope skipping intervention effectively improved EF in 12-14-year-old adolescents with overweight/obesity, with concurrent BMI reductions observed in the overweight/obese groups. Future research should focus on designing personalized exercise programs tailored to age and health status to optimize cognitive benefits. Furthermore, investigations should elucidate BMI's potential mediating role in exercise-induced EF improvements. Beyond phenomenological studies, mechanistic research should explore underlying pathways including neuroplasticity, gut microbiota, and inflammatory responses. Longitudinal follow-up studies are also imperative to evaluate the sustained cognitive effects of exercise interventions and identify determinants of long-term efficacy. (Page 15, lines 445-454)

Round 2

Reviewer 1 Report

Comments and Suggestions for Authors

Congratulations. The presentation of the manuscript was improved and I am recommending its acceptance.